# Contextual dependencies expand the re-usability of genetic inverters

Huseyin Tas [1,4], Lewis Grozinger[2,4], Ruud Stoof [2], Victor de Lorenzo [1✉] & Ángel Goñi-Moreno [2,3✉]

The implementation of Boolean logic circuits in cells have become a very active field within synthetic biology. Although these are mostly focussed on the genetic components alone, the context in which the circuit performs is crucial for its outcome. We characterise 20 genetic NOT logic gates in up to 7 bacterial-based contexts each, to generate 135 different functions. The contexts we focus on are combinations of four plasmid backbones and three hosts, two *Escherichia coli* and one *Pseudomonas putida* strains. Each gate shows seven different dynamic behaviours, depending on the context. That is, gates can be fine-tuned by changing only contextual parameters, thus improving the compatibility between gates. Finally, we analyse portability by measuring, scoring, and comparing gate performance across contexts. Rather than being a limitation, we argue that the effect of the genetic background on synthetic constructs expands functionality, and advocate for considering context as a fundamental design parameter.

[1] Systems Biology Department, Centro Nacional de Biotecnologia-CSIC, Campus de Cantoblanco, Madrid 28049, Spain. [2] School of Computing, Newcastle University, Newcastle Upon Tyne NE4 5TG, UK. [3] Centro de Biotecnología y Genómica de Plantas (CBGP, UPM-INIA), Universidad Politénica de Madrid (UPM), Instituto Nacional de Investigación y Tecnología Agraria y Alimentaria (INIA), Campus de Montegancedo-UPM, 28223 Pozuelo de Alarcón, Madrid, Spain. [4]These authors contributed equally: Huseyin Tas, Lewis Grozinger. ✉email: vdlorenzo@cnb.csic.es; angel.goni@upm.es

The abstraction of gene regulatory signals into on (high) and off (low) values allows for the design and implementation of genetic Boolean circuits[1] inspired by digital electronics. Such devices result from assembling two or more genetic logic gates[1,2]—the basic unit for processing information in genetic circuits based on Boolean logic. A core objective of synthetic biology[3] is the building of new regulatory circuits to compute inputs into outputs according to predefined logical functions[4], which are then used in a number of applications, ranging from bioproduction[5] to medical diagnosis[6]. Although this approach has been relatively successful, genetic logic gates are far more fragile and less reliable than their electronic counterparts as their signals are rarely constant and often fluctuate over time[7,8]. Consequently, the large-scale control of gene regulation based on Boolean logic alone is challenging. The central underlying issue is that a number of features intrinsic to biological systems, such as gene expression noise, analogue signalling[9] and evolutionary dynamics[10], make the intracellular environment an unsuitable domain for engineering idealised Boolean logic[11].

A fundamental challenge for the design of robust synthetic circuits, which underpins this work, is the oversimplified model that assumes DNA elements (i.e., gates) alone explain the performance of genetic circuits. Based on this assumption, the host chassis (the cell that receives a specific genetic construct) is generally ignored and the interplay of a genetic circuit with the host context is most often overlooked in the bottom-up design process—an issue that has been identified as essential for the

predictability of synthetic biology devices[12]. Our results here suggest that, rather than being antagonistic, incorporating context into the design of biological circuits can actually provide advantages by enlarging the available design space. Both the burden imposed by synthetic constructs on the host[13,14] and the impact of context on genetic activity[15], have phenotypic implications that cannot be predicted from a gene-centric standpoint. A common strategy seen in nature is to achieve a similar outcome using a different pathway in different organisms, rather than normalising pathways across all organism. For instance, E. coli solves energy requirements through the EMP metabolic pathway, while P. putida does it via the ED pathway. The function is the same: glucose as input and energy as output, but the circuitry is not normalised. Rather, it depends on the context. Recently, the term host-awareness[16,17] has been coined to bring attention to this problem, which is at the core of the lack of part interoperability[18] (i.e., parts that show similar performance in different host contexts). Here, we propose to utilise a strategy that is inspired by nature, and includes context as a parameter in the design of optimal genetic circuits.

While most synthetic biology efforts make use of only one host chassis to develop and characterise genetic constructs, potential applications may require the same genetic devices to work with different cell types[19]. For instance, circuit-constructs optimised in Escherichia coli for rapid prototyping, might be implanted into Pseudomonas putida for a bioremediation application[20] or into Geobacter sulfurreducens for bioelectricity production[21]. However, circuit performance will likely differ in different chassis, gene dosages and vectors, highlighting the importance of context in host-circuit design. As a result, the performance of a given genetic logic device would not only be a consequence of its DNA sequence but also would be influenced by its context. Within this scenario, modifying the context could fine-tune the performance of logic gates, thus engineering reconfigurable genetic logic devices which share the same sequences but exhibit different behaviours[22]. In the work presented below we inspect these scenarios by analysing quantitatively the behaviour of a collection of genetic inverters in different strains of the same species, in other species and in either case with the same devices borne by low, medium and high-copy-number vectors. The results illustrate that playing with these biological backgrounds expands the range of parameters that rule the behaviour of each construct. On this basis, we consider that context variability could be an advantage for circuit design rather than being seen as problematic.

## Results

**Generation of gate-context libraries**. To generate enough data on the contextual dependencies of genetic inverters we made use of 20 NOT logic gates assembled with a suite of promoters and repressors first developed as components of the CELLO platform for E. coli[1] and then recloned in broad host range vectors of different copy numbers for delivery to different types Gram-negative hosts[23]. The logic function (NOT or inverter) corresponds to a genetic device that reverses the incoming signal (i.e. output high to input low and vice versa). The inverters used are pairs of a specific regulator (repressor) and its cognate promoter (Fig. 1a). The characterised transfer functions measured the impact on promoter activity (output; captured by the expression level of an ypf reporter fused downstream) generated by specific concentration of regulator (input). In order to manipulate the expression level of the regulator, its coding sequence was placed under the control of a lac promoter, which was externally induced by IPTG. For gates characterisation, these were transformed into a bacterial host, which was then used to measure the NOT function (Fig. 1b). Relative promoter units (RPU) for both the

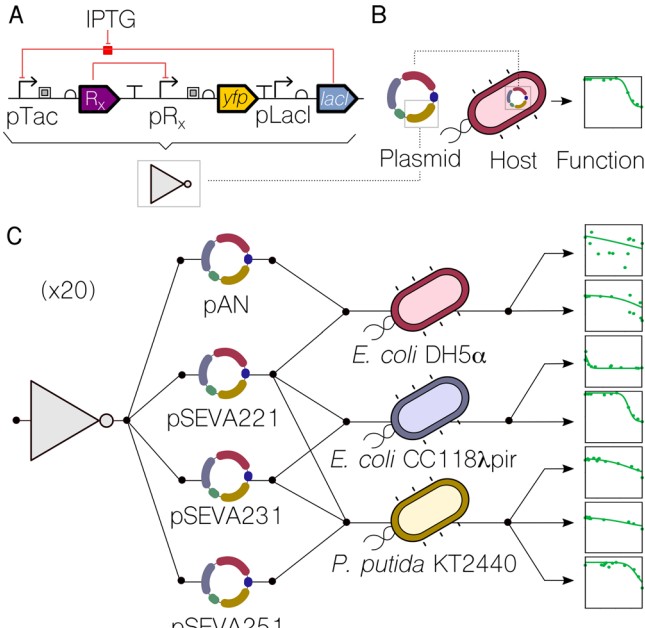

**Fig. 1 Generating a library of gate-context devices. a** Genetic inverters (NOT logic gates) were placed in between the pTac/LacI system (the input) and the yfp gene (the output). Key components are a repressor (R_x) and its cognate promoter (pR_x). **b** For a genetic construct to be measured, it needs to be cloned into a plasmid which, in turn, is transformed into a host cell—thus using a single context. **c** In this work, each genetic inverter (from an initial library of 20 gates) was measured in a number of context setups. These setups were based on combinations of two plasmid backbones (pAN and pSEVA), one of which with three different origins of replication—RK2 (221), pBBR1 (231), RFS1010 (251)—and three different hosts (E. coli DH5α, E. coli CC118λpir, P. putida KT2440). The performance of the resulting 135 gate-context devices was characterised experimentally by using flow cytometry and analysed computationally to find the impact of contextual dependencies on inverter's behaviour.

inputs and outputs of transfer functions were derived from yfp fluorescence measurements, in order to standardise their characterisations. The reference dataset of the behaviour of the 20 gates under inspection in *E. coli* NEB10β—12 main gates plus 8 variants—was retrieved from Nielsen et al.[1] (Table S1).

To assess the impact of the host context on gate performance, both the plasmid backbone and the cellular chassis were changed. As far as the carrier backbone is concerned, gates were cloned into the pAN and pSEVA[24] backbones, considering different origins of replication that led to low (RK2, pSEVA221), medium (pBBR1, pSEVA231) and high (RFS1010, pSEVA251) copy numbers. This contextual feature accounted for dynamics generated by circuit burden[25], since more copies of the same gate would increase the cost (of running it) to the cellular machinery. Regarding the chassis, we used two *Escherichia coli* (DH5α and CC118λpir) strains that are evolutionary relatively close and one *Pseudomonas putida* (KT2440) strain that is an evolutionary distant host to the other two. Combinations of these resulted in a library of gate-backbone-host devices (Fig. 1c) where the final performance cannot be explained by the genetics of the NOT logic gate alone. That is, the DNA sequence of the constructs is not enough to predict the behaviour of the gate— information about the context is then essential for understanding the genotype-to-phenotype dynamics. As shown in Fig. 1c, each logic gate in this study can have up to seven context-dependent dynamic behaviours, some of which differ significantly. Specifically, plots shown in Fig. 1c correspond to the characterisations of gate PhlF (one of the 20 gates of the initial library) in seven different contexts. While the performance changes abruptly in some cases (e.g., in contexts 3 and 4), it did not change significantly in others (e.g. in contexts 5 and 6). The codes and methods used for this analysis are made available to encourage extensions to this work and its application to other data sets (see Methods). Overall, our analysis suggests that contextual dependencies act as a hidden layer of parameters that must be carefully considered to achieve a predictable logic gate design—an issue which has been traditionally overlooked.

**Effects of cross-context portability.** When a genetic logic gate is either passed onto another organism, or carried by a different backbone, the interplay between itself and the context changes[26]. Contextual dependencies are adjusted. These modifications alter the expression levels of a gate, its dynamic range and (in some cases) its logic function. Moreover, context-dependent changes of qualitative behaviour imply that the dynamics of the interplay between context and construct are nonlinear. That is, a given pair of gates may suffer similar modifications in one context but very different in another.

For example, PsrA-R1 and PhlF-P2 show these effects (Fig. 2a). When both gates are hosted by chassis *E. coli* DH5α, the backbone (either pAN or pSEVA221) seems to play a key role in the logic outcome of PhlF-P2, which becomes more step-like with pSEVA221 (i.e. sharper transition from on to off). In contrast to this, gate PsrA-R1 does not follow that trend and remains qualitatively unchanged, although absolute expression values drop. Using the same backbone (pSEVA221) we then tested the context impact of varying the *E. coli* strain. Whilst the performance of PhlF-P2 is qualitatively the same (with smaller dynamic range), PsrA-R1 shows a qualitative change, becoming more step-like, thus showing more desirable behaviour than in other contexts. These inconsistencies in changes of qualitative behaviour of gates highlight the difficulty of compensating for such effects in order to engineer context-independent circuits[26]. However, there are also more predictable contextual changes in which that strategy may work well. For example, when both gates

are hosted by *E. coli* CC118λpir, changing the backbone from pSEVA221 to pSEVA231 (that only differ in the origin of replication) generates almost the exact same phenotypic modification. Finally, a marked difference occurs when the gates are hosted by *P. putida* KT2240. In these contexts, the gates lose their NOT logic, regardless of choice of backbone (pSEVA221, pSEVA231 and pSEVA251). The characterisation of the full library (20 gates) is shown in Supplementary Information Tables S4–23.

The issue of inter-context predictions arose as a formidable challenge. For example, in Fig. 2, an attempt to predict the performance that gates would display in the context *E. coli* DH5α (pSEVA221), based on gate performance in *E. coli* CC118λpir (pSEVA221) failed. The prediction was based on applying an optimised linear transformation to the gates transfer function. No linear transformation that performed consistently well could be found using this procedure, suggesting that a nonlinear transformation may be required. In this case, the optimisation was done using the AmtR-A1 gate, and the predictions tested on other gates in the library. As expected, some of the gates showed a relatively good prediction (good candidates for portability applications), but that was not the case for all of the constructs. Although predictable, gate portability is highlighted as an open problem and contextual dependencies offer a unique opportunity for fine-tuning gate performance, which we carefully analysed as explained next.

**Enhanced gate compatibility by fine-tuning contextual dependencies.** Building a genetic circuit by coupling genetic logic gates requires an assessment of their compatibility, to determine which gates can be connected. In order to connect two gates, the output levels of one must match the input levels for the other. If not, it may result in failure of the overall circuit logic[1,27,28]. This is one of the major bottlenecks that restrict the depth of genetic logic circuits and limit scalability, since not every gate within a library will be compatible with another. The analysis of inter-gate compatibility is therefore fundamental for circuit design and is an integral part of current synthetic biology Computer Aided Design tools[1,29]. However, knowledge about the effect of context on gate compatibility has until now been lacking.

In order to tackle this issue, we first scored the matching of all the gate pairs in the library according to their input and output thresholds (Fig. 3a). The inclusion of the input thresholds in the output ones defines a pair as "compatible". The extent of the inclusion is used to compute a compatibility score (formula presented in "Methods"), which is positive if the pairing is compatible and negative otherwise. This metric permits the comparison of all available compatible pairings and potentially informs design decisions. That is, under this framework, a design consisting of pairings with larger positive scores should be preferred over designs with comparatively smaller or negative scores. With this in mind, scoring of all pairings in a library may indicate the overall quality of a gate library and of the circuits produced thereof. Moreover, the information provided by the compatibility was complemented by the introduction of a similarity score (Fig. 3b). While the former relates two different gates, the latter relates the same gate to itself when varying contextual dependencies. This score quantifies the impact of specific context variations on each gate.

In this analysis, constructs were considered as gate-context entities (e.g. *E. coli* DH5α (pAN::PsrA-R1) or *E. coli* DH5α (pSEVA221::PsrA-R1), rather than individual gates alone (e.g. PsrA-R1) so that results account for the performance of a gate in a given context. We consider that high numbers of compatible pairs in a library are a desirable trait, and examine the impact of

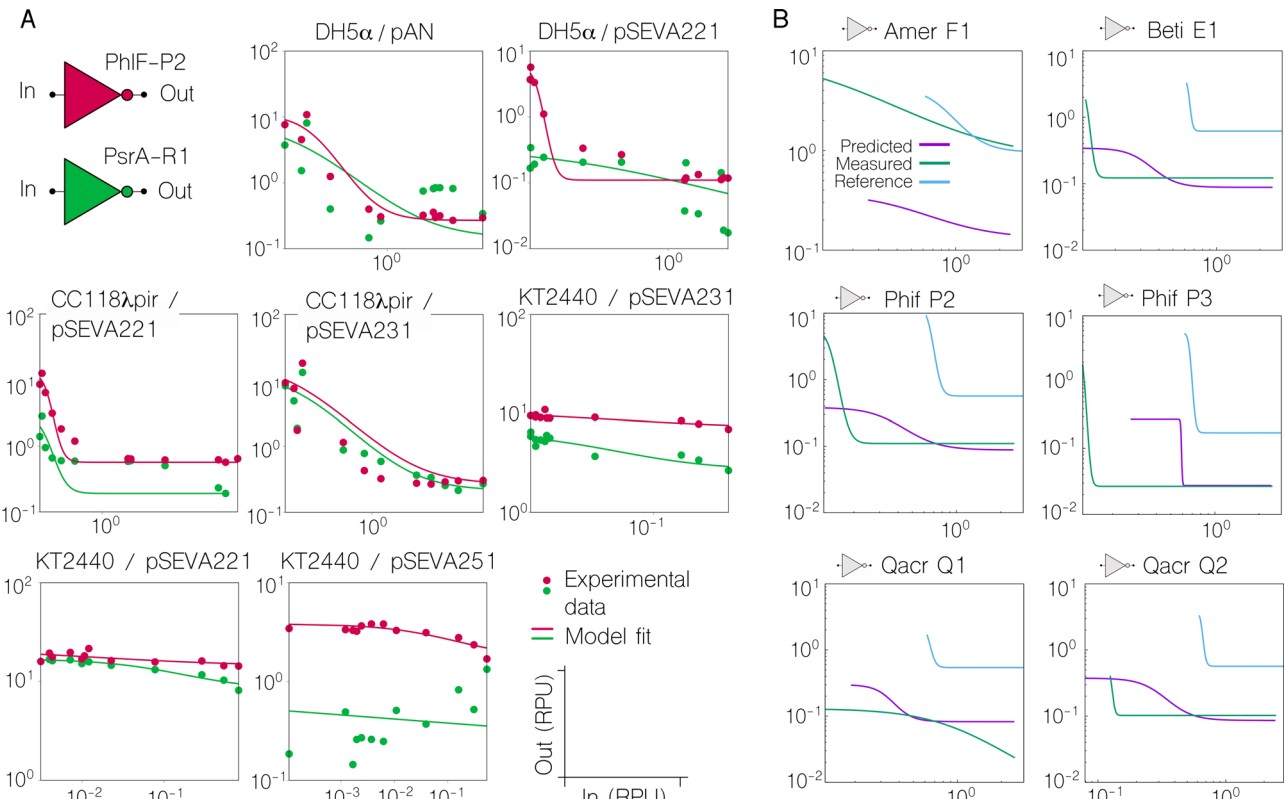

**Fig. 2 Nonlinear effects in the cross-context portability of inverters. a** Plots comparing the characterisation of two gates, PsrA-R1 and PhlF-P2, in different contexts. As well as each individual characterisation is differing across contexts, the relationship between the two characterisations also differs, depending on both strain and plasmid. This is, some contextual changes impact on a similar way on the performance of two inverters, while others impact on a different way—what we refer to as nonlinear modifications. **b** Nonlinearities made the prediction of gate performance changes between contexts an overarching challenge. Predictions were made for gates in the *E. coli* DH5α (pSEVA221) context ('Predicted' line), based on their characterisations in *E. coli* CC118λpir (pSEVA221) i.e. 'Reference' line. Predictions were made using a transformation matrix found by searching for the optimal linear transformation between the AmtR-A1 gates in each context. The actual characterisation of the gate is shown for comparison ('Measured' line). It can be seen that the optimised linear transformations cannot accurately predict changes in gate behaviour between contexts. In particular, although translations (a linear transformation) in the Input and Output axis appear to be predicted well in some cases (see for example QacR Q2), more qualitative changes in the shape of the response curve cannot be addressed by this linear transformation (see for example QacR-Q1). All response curves are plotted in RPU-RPU.

the two contextual features we focus on (backbone and host) on this metric, both independently and together. As a general trend, relaxation of the contextual parameters of backbone and host results in an increase in compatible pairings. More importantly, this increase is made up of not only new pairings within the additional contexts, but also additional pairings between gates in different contexts. In the example shown in Fig. 3c, 85 pairings are possible in DH5α without using two different backbones, 67 with pAN (Fig. 3c, left) and 18 with pSeva221 (Supplementary Figure S3), whereas 203 pairings are possible when allowing mixing of these backbones (Fig. 3c, middle). Thus, compatible pairings in the library increased ~240% as a result of incorporating connections between gates with different backbones. A similar jump of ~240% is observed when incorporating connections between gates with different hosts in addition to different backbones (Fig. 3c, right). We conclude that consideration of backbone and host as a design parameter results in a more flexible, and reconfigurable, library with the ability to include dynamics that are not captured by just DNA sequences e.g., the copy number of circuits (thus their burden to the cell).

Due to the consideration of contextual dependencies of host and backbone, the original gate library of only 20 genetic devices was increased to 135 different functions. Therefore, there were many more options to evaluate and more compatible pairs found. However, some of these pairs correspond to gates that are

compatible only if they are inside different hosts. For example, the gates HlyIIR-H1 and AmeR-F1 can only be matched (i.e., their function is complementary) if the former is hosted by *E. coli* DH5α and the latter by *E. coli* CC118λpir. This suggests that taking a multicellular (distributed) computing approach[30–33] will be required to couple the functions of these two constructs. In multicellular computations, a predefined function is distributed across different engineered bacterial strains (or species), which are connected in such a way that the output of one cell is the input of another one. Therefore, considering the host of a genetic construct within circuit design will allow for building both intra- and inter-cellular computations[11].

**Context-aware design rules for layered logic gates.** The design of synthetic genetic circuits typically overlooks contextual features by considering that phenotypic performance can be explained by the DNA sequence of the synthetic construct alone. However, this over-simplification has negative implications; for example, it requires considerable effort to adapt a genetic circuit to a new host[34]. The fact that genetic constructs show different dynamics depending on their context is not necessarily a disadvantage for predefined circuit design—could we rationally use such variability? To begin to address this question, we carried out computations in order to identify the maximum circuit depth

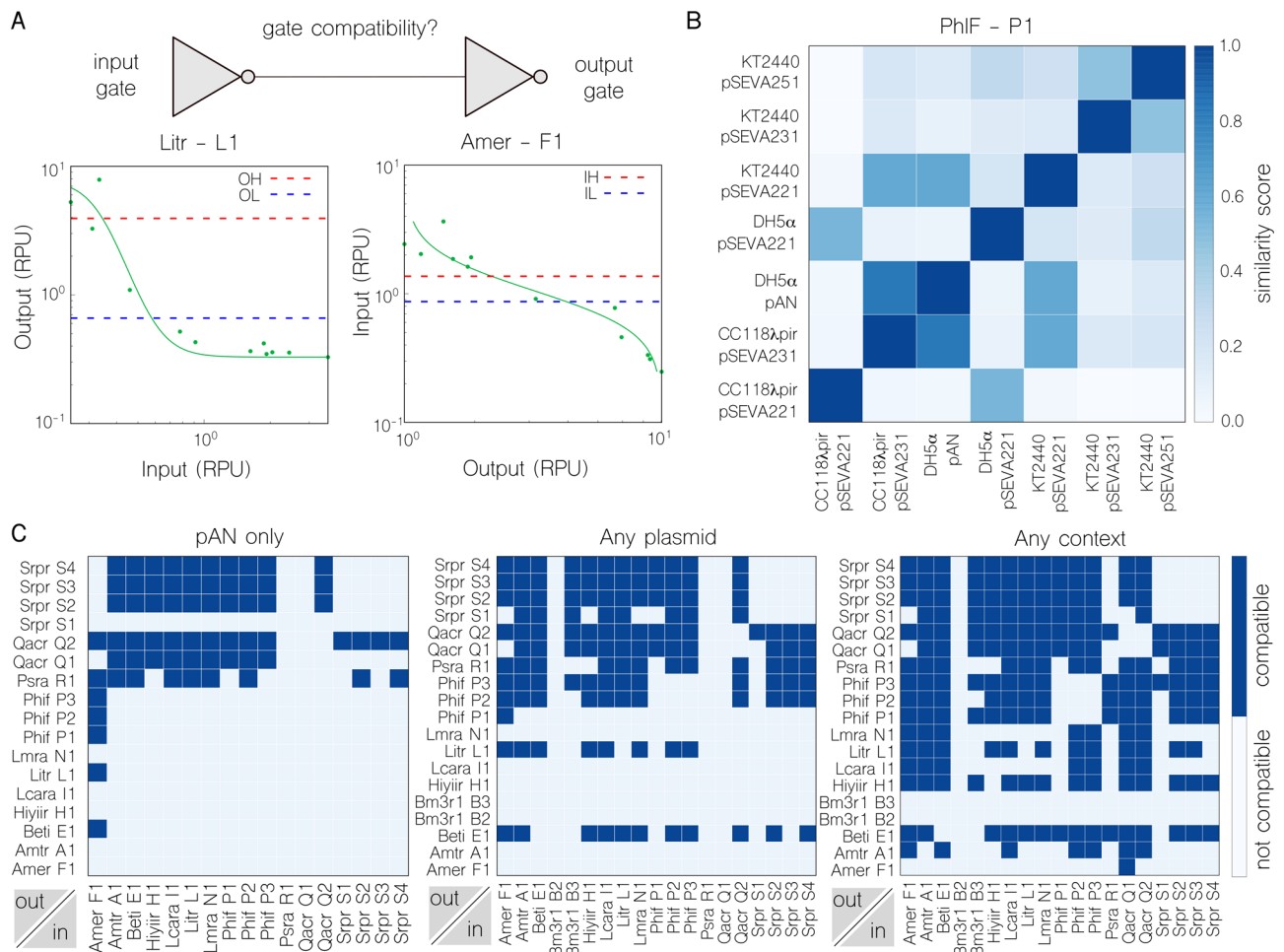

**Fig. 3 Comparing inverter compatibility and similarity across contexts. a** Gate compatibility indicates if two gates can be sequentially assembled—the output of the first gate is compatible with the input of the second—or not. Since the IH (Input High) and IL (Input Low) thresholds of the output gate, AmeR-F1, lie between the OH (Output High) and OL (Output Low) thresholds of the input gate, LitR-L1, this pairing is compatible. **b** A heatmap of similarity scores (which refers to how similar the shape of both inverter's transfer function is) calculated using discrete Frechet distance between the characterisation of PhlF-P1 in each of the seven contexts (darker is more similar). Most values within the score scale are covered, which highlights context contribution to final gate behaviour. **c** Maps of compatible pairs for the gates characterised in: the strain *E. coli* DH5α with pAN as the only plasmid for all inverters (left), the strain *E. coli* DH5α with any variation in plasmid type (middle) and in any context choice (right). Considerably more compatible pairs are found when freedom is given in the choice of backbone, rising from 67 (left) to 203 (middle) pairs. The freedom to use both backbone and strain as a design parameter yields the most compatible pairs at 697. Further, from 19 functional NOT gates, with a possible 320 pairings between them, 198 of these (61.8%) can be realised by allowing different backbones and strains to be utilised.

(i.e., number of layers35) that could be achieved by connecting gates within our library, and assessed the impact of contextual effects in such a chain (Fig. 4). First, when considering all gates in the same context, with backbone pSEVA221 and hosted by *E. coli* CC118λ*pir*, the maximum depth was 3 (Fig. 4a). That is, there are three gates that can be connected consecutively while maintaining the correct logic output (i.e., logic values 0/1 are effectively transmitted from beginning to end). Every other valid configuration will result in fewer (or the same) number of layers. We find that increasing the number of contexts available can significantly increase the maximum depth computed by the search algorithm. As shown in Fig. 4b, allowing another context by including gates characterised with any backbone (but still hosted by *E. coli* CC118λ*pir*) increases the maximum depth to 5. This can be further improved upon by allowing freedom in the choice of host, for a total of 7 contexts (Fig. 4c). In this case, the computed maximum depth is 11, far beyond the current state-of-the-art for synthetic circuitry1. Of course, this maximum circuit depth is a hard upper bound on the depth of any circuit that could be constructed using the library, but does not guarantee

that this depth can be achieved in a circuit that does not simply layer inverters. However, the increasing depth we observe as contextual parameters are relaxed suggests that there is potential for deeper circuits with context as a design parameter than without. Libraries of genetic gates which are allowed to be placed in multiple contexts appear to be less restrictive than their single-context counterparts, and could potentially permit a broader range of more complex circuit designs.

## Discussion

A fundamental driving force for synthetic biology[3,35] is the clarification of mechanistic assumptions as our understanding of molecular processes increases, which allows scientists to add novel tools to the catalogue for engineering living systems. Although the cellular environment consists of much more than DNA, circuit design[1,29] typically revolves around genetic elements (promoters, terminators, RBSs…) in order to link genotype to phenotype—an oversimplified reductionist approach. The comfortable, yet error-prone, assumption that engineered parts

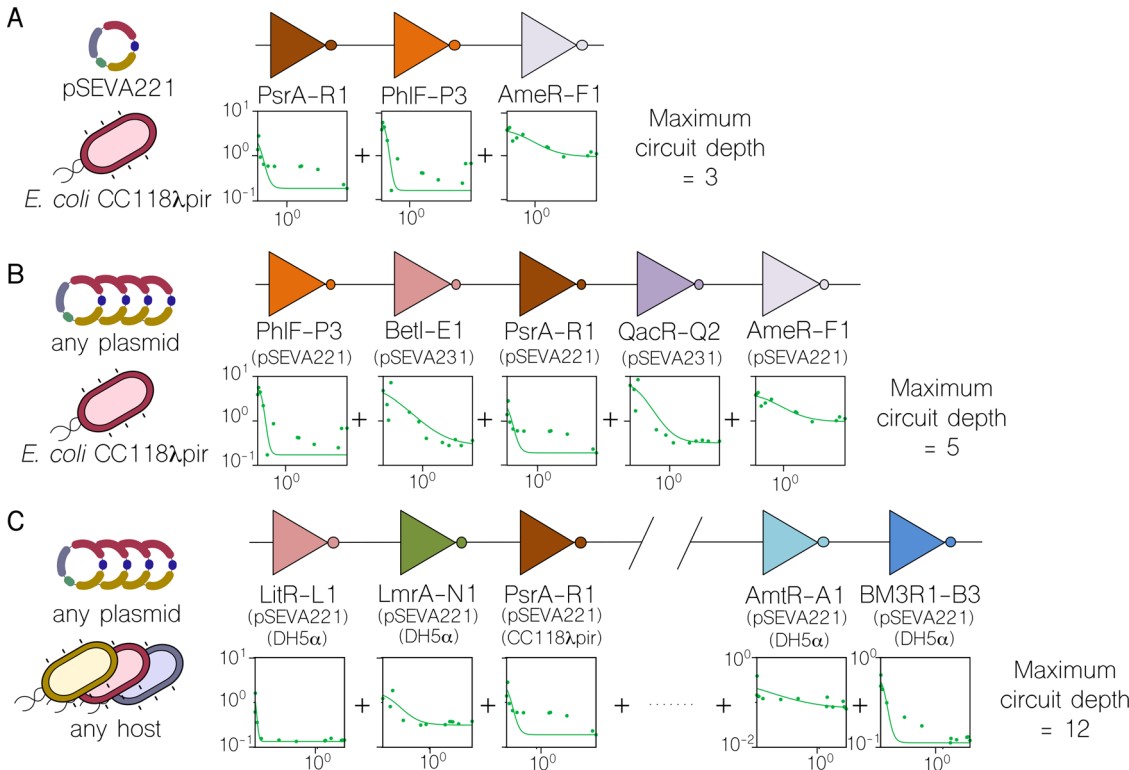

**Fig. 4 Calculation of maximum circuit depth as a result of layering inverters.** Based on the compatibility between gates, these were layered within the library in order to evaluate the impact of contextual dependencies on circuit size. For all graphs: x-axis refers to the input and y-axis to the output (both RPU). **a** The maximum depth calculated when the computational method is forced to consider all gates carried by the low copy-number plasmid pSEVA221 and hosted by Escherichia coli CC118λpir, is 3 gates-deep. **b** If the algorithm is free to select any plasmid (but still forced to E. coli CC118λpir), the maximum depth increases to 5. In this scenario, two gates are carried by the medium copy-number plasmid pSEVA231. **c** In the last analysis, the calculation used all contextual dependencies, including the variation in host chassis. The maximum number of gates layered increases to 12 (only 5 shown in figure—refer to Supplementary Material for more information). In the sketch shown in the figure, 4 out of the 5 gates were characterised in the strain Escherichia coli DH5α.

alone can ultimately explain phenotypic performance needs to be expanded upon[12]. This leads us to consider what has been termed genetic background[15] and host-aware[16] dynamics: cellular features and constraints that have an impact on circuit performance but are not captured by the DNA sequences of the construct. In recent years, several of these features have been analysed: the impact of having limited cellular resources[36,37] (e.g. ribosomes) to "spend" on synthetic constructs, the effects of placing DNA parts in different genomic locations[38,39], the role played by metabolism in genetic control[40,41], or even genetic stability[42] due to evolution over time. All these effects turn the portability of genetic circuits into an overarching challenge—the fine-tuning of a circuit to work inside a different host (to the one it was originally built-in) is still a major task[26]. Furthermore, it limits the scope of biological circuits by solely using a DNA-insert toolbox for designing circuits.

Here, we use the word "context" to refer to the molecular background of the cell beyond genes and analyse how such context can be used for improving biocircuit design. By "dependencies", we mean the constraints imposed by the context on a given genetic construct. Therefore, genetic logic gates are exposed to contextual dependencies that influence their phenotypic behaviour. In the extreme case, these constraints can even result in unviability of the cellular host, for example, during the conversion to pSEVA broad host range backbones, five of the genetic inverters were not functional when cloned into a high-copy plasmid (pSEVA251), perhaps due to overload in the allocation of cellular resources resulting in toxicity[23].

Although synthetic biology is a field full of metaphors[43] already, we entertain here a new one that we consider to provide a useful conceptual frame: the use of contextual dependencies as in a software engineering problem. Any piece of software, or program, must run inside a specific environment (e.g. operating system) and software engineers usually face the problem of adapting it to the particular dependencies of the environment/context at stake. Under this metaphor, genetic circuits are considered software (instead of hardware[44]) whose performance is deeply linked to context-specific dependencies, which can allow designers to access functions that could not be coded otherwise. In this paper, we propose that contextual dependencies are important parameters for circuit design, and focus on [i] backbone carrying the construct, and [ii] cellular host in which the construct performs.

In this work, we exploited a library of 20 genetic inverters (NOT logic gates), which are combined with four different backbones and three cellular strains to give a total of 135 gate-context constructs. In this regard, the number of functions exposed by the library increases by 675% due to the addition of these two contextual dependencies. With this new library we carried out experiments in order to assess the implications of adding context to the context-free initial collection of NOT gates. First, the characterisation of the constructs showed how gate behaviour changed across contexts in a nonlinear fashion. That is, the phenotypic modifications in the performance of one gate across two contexts may not match those of another gate under the same contextual transformations. This has major implications

for the portability of genetic devices, since not all genetic components may be affected in the same way upon host change—thus building complex portable devices will become difficult (if not entirely impossible). Second, our experiments suggested that the compatibility of gates (so that they could be composed; the output of the first being the input of the second) does not only depend on selected genetic inserts, but also on their context. While only 67 compatible pairs were found in the original library of 20 inverters, the number increased to 697 in the new library. For instance, by allowing gates to be carried by four different backbones, the computational algorithm was able to evaluate the compatibility of four functions instead of 1 and return not only the name of the compatible gate but also the name of the backbone to use for carrying it. This allows reconfiguration of genetic constructs, since the same piece of DNA-insert can have different behaviours depending on rationally selected contextual dependencies. Finally, the use of the cellular host as a separate design parameter allowed identification of gate pairs that were only compatible if connected gates were located in different strains/species. Consideration of different contextual dependencies was found to increase the theoretical maximum circuit depth from 3 to 12, as shown in Fig. 4. Practical implementations of such context-dependent designs would comply with the following three rules. Firstly, designs necessarily rely on multicellular distributed computing approaches[30,32,45,46] in order to connect logic gates in different hosts. These connections could be established by using orthogonal quorum sensing (QS) systems[47,48]. In order to prevent the number of orthogonal QS systems from being a limiting factor, paths should be selected that minimise these inter-host connections—therefore maximising the number of gates per host. Secondly, linking gates inside a host requires using repressor molecules for signalling, which must also be orthogonal to ensure correct gate operation. Since the library of repressors would also be limited, the adoption of multicellular approaches offers an important advantage: the re-usability of parts i.e. a repressor that is used in one host may be reused in another. This second rule may be used to distribute circuit burden across different strains. Finally, plasmid backbones can coexist inside the same host as long as their origins of replication are different—otherwise, some plasmids may be lost during the process. Altogether, these guidelines establish a rational criteria for the selection of context-dependent circuitry components from a bottom-up design.

In a similar way to living systems that use a number of mechanisms to go from genotype to phenotype, we advocate for the development of genetic circuits by considering whole-cell dynamics—including contextual dependencies. Although in this work we have considered backbones and strains as the contextual dependencies, this library could be extended by adding other contextual parameters such as other promoters, context-specific genetic parts, or substrates. This will result in the design of biological circuits that are closer to the internal workings of natural systems[11]—therefore more robust, reliable, predictable and reproducible.

## Methods

**DNA and strain construction.** All cloning steps were done in E. coli CC118λpir. Primers are ordered from Merck Sigma Aldrich, Inc. The repressible and inducible systems were previously described by Voigt Lab and acquired by the courtesy of Voigt Laboratory in MIT (USA). Description of the 20 different NOT gates moved into broad host range pSEVA backbones are described in Tas et al.[23]. Components of the original inverters, like terminators, RBSs and insulators were kept the same during the SEVA conversion. SEVA backbones have two terminators, T0 and T1 which are important to lower potential leakages. Required oligo list can be found in the Supplementary Information (Supplementary Table 3). The pAN backbone[1] has

a kanamycin resistance gene with a p15A origin of replication which is ~15 copy number in E. coli NEB10β strain.

**Medium and experimental protocols.** In all experiments (unless stated otherwise) M9 minimal medium for E. coli and M9 medium for P. putida were used. The ingredients of the M9 medium used are as following: for 250 ml of liquid medium, 25 ml 10X M9 salts, 500 μl of 1 M $MgSO_4$, 2.2 ml of 20% carbon source (glucose for E. coli and citrate for P. putida), 125 μl of 1% Thiamine, 2.5 ml of 1% Casamino acids and milliQ-$H_2O$ up to 250 ml. The concentration of kanamycin used is 50 μg ml$^{-1}$ in the experimentation procedures. IPTG was used as inducer for pTac/LacI inducible system in 12 different concentrations diluted from 1 M stock concentration that are 0, 5, 10, 20, 30, 40, 50, 70, 100, 150, 200, 500 and 1000 μM. For synchronising the cells in the experimental procedure, cultures are started from a single colony picked from LB agar plate which is each time freshly prepared from −80C glycerol stock by inoculating it O/N in 1 ml M9 minimal medium. O/N cultures after saturation were diluted by ~666 times to inoculate 200 μl M9 minimal medium in 96 well plate for 24 h, which is enough to reach to 0.2 - 0.3 OD in 96 well plate after which for halting the growth cells were kept on cold platform during the measurements.

**Flow cytometry analysis.** Miltenyi Biotec MACS flow cytometer at channel B1 with an excitation of 488 nm and emission of 525/50 nm was used for measuring YFP fluorescence distribution of each sample. 30000 events were defined as the statistically sufficient amount under singlet gating for each sample. Calibration of the flow cytometer was done daily by using MACSQuant Calibration Beads. Throughout flow cytometer measurements samples were always kept on cold 96 well plate platforms. For the analysis of the data FlowJo software was used. In the analysis, gating was done via the usage of auto-option and allowing to cover at least 50% of the whole events run while Forward and Side scatters were plotted, and the same gating conditions were kept for all samples in the same group.

**Fluorescence data pre-filtered by cell size.** In order to unify fluorescence measures between and within flow cytometry experiments, we analysed fluorescence and scattering values. Variation in cell size across experiments showed that median fluorescence values were decisively affected, therefore inaccurate for the sake of comparison. To compare between experiments, we took the distribution of fluorescence for single scattering values. A full description of this process is detailed in Supplementary Information.

**Standard fluorescence measurements.** Two extra plasmids were used for measurements, the autofluorescence plasmid (Backbone::1201), and the reference standard plasmid (Backbone::1717) that triggers *yfp* expression under the pLacI constitutive promoter. In order to derive relative promoter units (RPU), the following equation was applied:

$$\text{RPU} = <\text{YFP}> - \frac{<\text{YFP}>_{\text{autofluorescence}}}{<\text{YFP}>_{\text{standardization}} - <\text{YFP}>_{\text{autofluorescence}}},$$

where <YPF> stands for the median fluorescence value of the inverter that is to be standardised into RPU, <YPF>$_{\text{autofluorescence}}$ is the median fluorescence value of the autofluorescence plasmid, <YPF>$_{\text{standardisation}}$ indicates the median fluorescence value from the standardisation plasmid. RPU values were calculated in transfer function plots for different data points using at least 6 inducer levels covering the range of induction up to saturation.

**Data fitting.** The pre-filtered experimental data were fitted to a 4-parameter hill equation of the form

$$h(x) = y_{\min} + \frac{(y_{\max} - y_{\min})k^n}{k^n + x^n}.$$

This model is often used to relate gene expression levels to the concentration of a repressive transcription factor, most notably in[1], where it is used to describe the input–output relationship of genetic inverters (NOT gates). In this study, we consider only NOT gates, for other types of gate, other models should be selected

The parameter values for $y_{\min}$ and $y_{\max}$ were set to the minimum and maximum of the corrected experimental data. The values for $k$ and $n$ were then fitted using the least squares method from the scipy.optimize Python package[49], with logarithmic residuals.

Fits were obtained for all gates presented in this paper. The fitted parameters are shown in tables in the Supplementary Materials. However, for many gates, it was found that the model was a poor fit to the data. In this case, the gate cannot be considered an example of a NOT gate and should not be used as parts in circuit design. In particular, the criteria for valid inverters as described below should omit these poorly behaved gates.

**Calculating compatibility.** Thresholds OL, OH, IL and IH were computed from the parameters of the fitted hill curves according to the definitions given in [1]. OL and OH are twice $y_{\min}$ and half of $y_{\max}$, respectively. IL and IH are the values of $x$

for which the output of the fitted hill function is equal to OL and OH, respectively. Accordingly, the values of IL and IH were calculated with the following formulae:

$$\text{IL} = \left(\frac{k^n y_{\max}}{(y_{\max} - 2y_{\min})}\right)^{1/n},$$

$$\text{IH} = \left(\frac{k^n (y_{\max} - 2y_{\min})}{y_{\min}}\right)^{1/n}.$$

Inverters were considered operational under the condition that OH>OL and IH>IL for their fitted hill curve.

**Compatibility scoring**. For a pair of operational inverters, A and B, their compatibility score was defined as

$$\min\left(\ln\left(\frac{\text{IL}_B}{\text{OL}_A}\right), \ln\left(\frac{\text{OH}_A}{\text{IH}_B}\right)\right).$$

A positive score indicates that a high(low) output from A will also be interpreted as high(low) by B, because IL(IH) of B is greater(less) than OL(OH) of A. From this we imply that inverter A can be connected as input to inverter B, if and only if their compatibility score is positive.

**Computation of inverter chains**. Chains of compatible inverters were found by creating a table of compatibility between available inverters, for which the entry for a compatible pair was 1, and all other entries were 0. This table was then treated as the adjacency matrix of the graph of all possible connections, and the longest paths were enumerated using a depth-first search of the graph. Paths in which the same repressor was used more than once were excluded from the results, thus imposing an upper bound of 12 on path length.

**Similarity measure**. The discrete Frechet distance[50] was used to measure similarity of the shapes of two experimental curves, after first log transforming and min-max normalisation of the data along both axes. The Frechet distance was then subtracted from 1 in order to produce a metric that increases as the shape of the curves becomes more similar. The discrete Frechet distance was computed using the 'similarity measures' Python package[51].

**Prediction**. The goal of the prediction is to transform the characterisation of gates in a source context, to a characterisation in the target context. A single operable gate was selected arbitrarily upon which to base the prediction. The 'scipy.optimise' Python package[49] is used to compute a linear transformation matrix, which when applied to the source characterisation, minimises the L1Loss between the transformed characterisation and the target's true characterisation. Predictions for other gates in the library are then made by applying the same transformation to their characterisation in the source context.

**Reporting summary**. Further information on research design is available in the Nature Research Reporting Summary linked to this article.

## Data availability

Availability of data, genetic material and supporting software. The flow cytometry data used for analysis in this study is available as a figshare repository at https://data.ncl.ac.uk/ndownloader/articles/12073479/versions/1. This SBOL files for the genetic constructs used in the study are available at https://github.com/lgrozinger/pyolin/tree/master/results/sbol. The constructs themselves are retained at SEVA bank (http://seva-plasmids.com) at CNB-CSIC, Madrid, Spain and ready for distribution for research purposes.

## Code availability

The Python package used to perform all the analysis presented, the preprocessing of the raw cytometry data, and to generate the figures shown here, is made available at https://github.com/lgrozinger/pyolin.

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

## Acknowledgements

Authors are indebted to Esteban Martinez-Garcia for his feedback on this Project. This work was funded by the *SETH* (RTI2018-095584-B-C42) (MINECO/FEDER), SYCOLIM (PCI2019-111859-2 ERA-COBIOTECH 2018) Project of the Spanish Ministry of Science and Innovation. MADONNA (H2020-FET-OPEN-RIA-2017-1-766975), BioRoboost (H2020-NMBP-BIO-CSA-2018/ 820699), SYNBIO4FLAV (H2020-NMBP/0500) and MIX-UP (H2020-Grant 870294) Contracts of the European Union, the InGEMICS-CM (S2017/BMD-3691) Project of the Comunidad de Madrid - European Structural and Investment Funds - (FSE, FECER), and the SynBio3D project of the UK Engineering and Physical Sciences Research Council (EP/R019002/1). G-M was also supported by grants from Comunidad de Madrid (Atracción de Talento Program, grant 2019-T1/BIO-14053) and the Severo Ochoa Program for Centres of Excellence in R&D from the Agencia Estatal de Investigación of Spain, grant SEV-2016-0672 (2017-2021).

## Author contributions

V.D.L. and A.G.M. conceived the study and wrote the article. H.T. carried out the experimental part of the work. L.G. carried out the computational side of the study. H.T. and L.G. performed data analysis. All authors contributed to the discussion of the research, interpretation of the data and manuscript edition.

## Competing interests

The authors declare no competing interests.
