## [Peer Review File · Nature Communications]

Reviewers' Comments:

Reviewer #1:

Remarks to the Author:

In this study, Huseyin et. al. claim to address the problem of interoperability by interrogating the role of context – which is defined as the host cell plus the plasmid backbone – on the performance of genetic logic gates. In this work, they specifically examined 20 genetic variations of the NOT gate in different bacterial contexts. The authors claim their results demonstrate that each gate may or may not behave differently, depending on the context. Even if the pattern of change between two different contexts for one gate was known, they found it was sometimes difficult to predict the change in behavior between the same two contexts for a different gate, which the authors refer to as a "non-linear pattern". The authors went on to define a parameter for gate compatibility, which quantifies the likelihood that any two gates could be functionally connected in sequence. The authors used this parameter for gate compatibility to demonstrate that if we consider the idea of connecting gates from different contexts, the total number of compatible gates increases. In other words, one possible consequence of considering context is the potential to design circuits based on "distributed computing", where genetic gates may be housed in different plasmid backbones or host cells and are connected via external signaling molecules. The authors demonstrate with a simulation that if they were to use this approach, they would theoretically be able to achieve a much greater layer depth (i.e., the number of gates one could connect in sequence) than if they were constrained to using the same context.

Overall, the experiments in this study were sound. However, the notion that which plasmid backbone or the host is used matters is not a new idea per se, but is appreciated not just in synthetic biology but across many disciplines. That is one major reason why drugs/diagnostics are tested in multiple orthogonal animal models, because they could just work in one narrow system. Without demonstration of an important biological application, or interesting experimental validation of their model, this study feels lacking and does not represent a significant advancement for the field. The authors propose that one downstream consequence for considering gate-context is distributed computing and the potential to increase the number of layer circuits (up to 11 layers). However, no attempt was made to validate this prediction, which again makes the study feel lacking.

Major Concerns

1. One potential way to strengthen the impact of this work would be to experimentally validate their model (Fig. 4). These experiments might include something like (1) connecting NOT gates between two plasmids, (2) between two cells, and ultimately demonstrating (3) a circuit with depth = 11, which as the authors say, would be "far beyond the current state-of-the-art for synthetic circuitry" (page 7, line 211–212). These experiments would also require further support, including a comparative assessment of host cell growth rates (as stressed in ref. 47), demonstrating that all signaling molecules between cells and plasmids are orthogonal, etc.
2. The authors cite several papers to support the notion that distributed cellular computing is an active field and therefore a viable solution. However, there are significant limitations of this approach, which are not discussed in the manuscript (refs. 30, 32, 46, 47). The authors should add a more nuanced discussion of the drawbacks of distributed computing (e.g., differences in cell-to-cell behavior, orthogonality of signaling molecules, etc.).
3. The authors use the term "non-linear patterns" to describe the idea that the change in behavior for one gate between two different contexts may or may not map onto the change in behavior for another gate between the same two contexts. The authors should provide further justification for using this mathematical term or consider replacing it.

Minor Concerns

1. On page 5, line 139 there is a typo – "non-linear patters"

2. On page 3, line 80-82 – this description of the NOT function should be revised for clarity. The use of double negatives (e.g., "...not negatively regulated...") may be confusing to some readers.
3. From the methods, it appears that the Input is the IPTG regulator and the Output is the YFP reporter, which are quantified in author-defined units of Reference Promoter Units (RPU). These labels are missing completely from Fig. 1, Fig. 2, and Fig. 4. While Fig. 3 does show units of RPU, it does not label the identity of the molecule being quantified (i.e., IPTG, YFP). These labels should be added to all figures and mentioned in the captions.
4. Page 3, line 95 – the authors should add further clarification on why these specific cell lines were chosen, and why a larger and more diverse library was not used.
5. In multiple places (e.g., page 2, line 49-50; page 4, line 108) the authors claim that the role of context in genetic circuits is traditionally overlooked. They should add citations to support this claim.
6. The gate-compatibility score seems useful because it could be applied to any Boolean gate in any system, and is particularly useful for biological gates because HIGH/LOW values are not universally defined. I'm not sure why this is buried in the supplement; the authors could consider bringing this formulation to the main text with explanation.
7. Page 6 line 181-182 – The authors mention that when considering combining gates from different contexts, the total number of compatible gates (X) increases, but so does the number of possible gates (Y). Importantly, they mention that the percent of total possible combinations remains constant (i.e., normalize X/Y). This normalized percent should be represented in the figure for all three panels (i.e., pAN only, Any plasmid, Any Context) to avoid misleading the readers.
8. In Fig. 3C, why was pAN chosen as the single context to isolate in the main figures? It appears that from the supplemental figures, all contexts were tested, and that some imposed almost no compatibility constraints (e.g., DH5a host with the pSeva221 backbone.) Further, why was the DH5a host fixed for the "Any plasmid" table? Were other hosts not tested in this same manner? For completeness, the authors should also fix each plasmid and enable connections between different host cells. Were these specific isolations/contexts chosen simply to make the data show the trend that releasing constraints on gate context increases the number of total compatible gates?
9. The compatibility score is a continuous value and is reported as such in the supplemental figures. In Fig. 3, these values are binarized. How was this threshold chosen? If it was that positive scores are defined as compatible, how did you differentiate gates that are very positive from gates that are very close to 0? It is easy to imagine that with biological variance the latter gates would be subject to failing.

Reviewer #2:

Remarks to the Author:

This paper addresses the important topic of host context dependence in genetic circuit elements. Particularly, genetic circuit elements described using Boolean algebra that are intended to be composable into large functions. It focuses on NOT gates used in 7 bacterial based contexts. The key elements that need to be demonstrated in my opinion for this type of work are:

1. Reproducibility – this appears to be fine in this case from an experimental standpoint. In particular, a strength is the fact that their approach uses existing gate "architectures" from the "Cello" work. Naturally, if the author's work is also able to be automated, details on an automation framework to provide this analysis would be great. This would include capturing the protocols in frameworks like Autoprotocol, Aquarium, Protocols.io, etc. If automation is used, scripts for robots (e.g. PyHamilton scripts or even CSV picklist files), would be great.

2. Performance metrics – an interesting phenomenon of this work is that the NOT gates can have different functions in different contexts. A way to illustrate which Boolean algebraic function the circuits can implement could be shown in a "cosine similarity" style analysis like in ["Large-scale design of robust genetic circuits with multiple inputs and outputs for mammalian cells" – Wilson Wong]. The provided data fitting and compatibility analysis equations are adequate for this work.

3. Standardized measurement – It would have been nice to see something like MEFLs [“Quantification of bacterial fluorescence using independent calibrants” – Jacob Beal] used in the measurement work but the authors to their credit do work with a version of the standardized fluorescence measurements.

4. Availability of the genetic information and associated data – it is not clear that this work deposited the material in a location that can be retrieved. A suggestion is to create SBOL files for this work and store them in a public SynBioHub instance. As for the genetic material, it would be nice to know how to get these (e.g. Addgene). If those resources are not available to the authors, any proposal for how to accomplish this would be appreciated.

5. A method for others to use this data – while the tables and diagrams are great, this work really would lend itself toward a very simple design tool. The user could either enter the desired transfer function (using the functions specified in the paper) and/or host context. The tool then would present the viable options solely based on the data already calculated.

In the absence of a program, a clever way to use the heat maps along with transfer functions would be nice.

As it currently stands, there is no clearly accessible mechanism to make use of this data programmatically. This is a criticism of almost all work in this space. However, this work is very close to addressing this.

Python code for the data analysis work is provided (a strength), perhaps this could be improved to provide some of this design functionality.

6. Design rules that formally encode the context data discovered in this work – this work does address some of these elements with the compatibility matrices. However, these are not formalized. Again, if the authors were so inclined these can be written as a formalized set of mathematical rules related to composition. These rules could be written in python and used for a circuit design tool.

Another aspect that is not clear, is the role of other effects (e.g. toxicity) on the ability of the circuits to be composed. There are examples of the number of layers available but it is unclear how realistic these are. That might not be the point, but in the case of a paper that is trying to take broadly applicable ideas (gate composition) and make them realistic (host-context), ideas like this are a noticeable omission.

General Comments:

Why stop at 7 bacterial contexts or the fixed NOR gate architecture? It might be nice to add a couple of others (contexts like Geobacter or NOR gates using other promoter/repressor combinations) that are chosen AFTER you have done the initial analysis to confirm or reject predictions or to see how broadly applicable the lessons learned are.

I did not see the transfer functions for all 135 gates anywhere including the supplemental. This would be valuable even in the form of spreadsheet data.

Again the gate compatibility analysis screams out for a small tool. It is not clear from the supplemental what the python work includes. In general, it could be made more clear in the supplemental examples of what the software can do.

A natural next step now would be to introduce genetic elements that help to normalize the performance across context. Some discussion of this would be ideal in the manuscript.

I would expect the supplemental to have all of the gate information related to Figure 2 along with the actual numerical scores. Perhaps this is covered by the Python package?

In general, the supplemental material should be expanded with more data if nothing else for reproducibility checks across labs interested in this work.

In general, layering inverters is not particularly useful UNLESS you can produce different inverter behaviors with an odd number of inverters not possible by individual inverters or other collections of odd inverters. Understanding if new behaviors can be introduced with the layered gates would have been interesting.

Response to reviewers' comments to manuscript NCOMMS-20-28880

(answers in blue)

Reviewer #1

In this study, Huseyin et. al. claim to address the problem of interoperability by interrogating the role of context – which is defined as the host cell plus the plasmid backbone – on the performance of genetic logic gates. In this work, they specifically examined 20 genetic variations of the NOT gate in different bacterial contexts. The authors claim their results demonstrate that each gate may or may not behave differently, depending on the context. Even if the pattern of change between two different contexts for one gate was known, they found it was sometimes difficult to predict the change in behavior between the same two contexts for a different gate, which the authors refer to as a "non-linear pattern". The authors went on to define a parameter for gate compatibility, which quantifies the likelihood that any two gates could be functionally connected in sequence. The authors used this parameter for gate compatibility to demonstrate that if we consider the idea of connecting gates from different contexts, the total number of compatible gates increases.

This summarizes well the output of the manuscript.

In other words, one possible consequence of considering context is the potential to design circuits based on “distributed computing”, where genetic gates may be housed in different plasmid backbones or host cells and are connected via external signaling molecules. The authors demonstrate with a simulation that if they were to use this approach, they would theoretically be able to achieve a much greater layer depth (i.e., the number of gates one could connect in sequence) than if they were constrained to using the same context.

We refer to distributed computing approaches when multiple hosts are used as context (i.e., different gates in different hosts). In this scenario, the circuit must be assembled (or the logic gates must be layered) by using population-based methods. Since such methods are currently routine, we only refer to those as a way to highlight consortia for building yet increasingly complex devices. However, changing plasmid backbones alone also increases the complexity and it does not require any disturbed approach.

Something that must be clarified (we clarified this in the manuscript also) is that we do not demonstrate through simulations, as the reviewer remarks. We do not perform simulations, that would need to be experimentally validated. What we do is to assess the compatibility based solely on experimental data. Therefore, the number of gates that could connect in sequence is not the result of a simulation, but the consequence of analyzing gates' in-vivo performance. Such an increase in compatibility, by using contextual dependencies for design, is the target of the manuscript. It would be of interest to carry out design-oriented simulations and their corresponding validation; however, that will be the target of future efforts.

Overall, the experiments in this study were sound. However, the notion that which plasmid backbone or the host is used matters is not a new idea per se, but is appreciated not just in synthetic biology but across many disciplines. That is one major reason why drugs/diagnostics are tested in multiple orthogonal animal models, because they could just work in one narrow system. Without demonstration of an important biological application, or interesting experimental validation of their model, this study feels lacking and does not represent a significant advancement for the field.

While we understand Reviewer's concerns, we believe that they go beyond the scope of this ms. The fact that contextual dependencies expand the *reusability* of components is, to the very best of our understanding, a significant advancement for the field. Currently, when a set of DNA components (e.g., gates) does not behave as expected, or is not compatible with the previous/next device of the circuit, the most common practice is to replace it. However, the relatively few number of components that are available is a limiting factor that undermines the scalability of synbio devices. With our approach, there is no need to change the components, but their context, in order to modify functionality. Besides, this launches a more, let's say, bio-friendly way of bio-engineering, since it exploits a natural feature (as the reviewer acknowledges). We provide a library of component-context devices,

that we are convinced has a very positive impact in our field. And we perform an in-depth analysis on experimental behaviour to showcase the increase of compatibility due to contextual dependencies.

The authors propose that one downstream consequence for considering gate-context is distributed computing and the potential to increase the number of layer circuits (up to 11 layers). However, no attempt was made to validate this prediction, which again makes the study feel lacking.

As stated before, the reference to distributed computing is employed to justify the use of cellular host as a contextual parameter. That is, since multicellular approaches are available, there is no need to rule out chassis effect and build single-cell colonies. We analyse compatibility based on individual performance, and provide the results. Results that offer positive conclusions on the use of chassis as a parameter for layering circuits. The use of that results for design automation efforts would be the target of future work, and has always been out of the scope for this project. Finally, as we mentioned before, it is not a model/simulation prediction.

Major Concerns

1. One potential way to strengthen the impact of this work would be to experimentally validate their model (Fig. 4). These experiments might include something like (1) connecting NOT gates between two plasmids, (2) between two cells, and ultimately demonstrating (3) a circuit with depth = 11, which as the authors say, would be "far beyond the current state-of-the-art for synthetic circuitry" (page 7, line 211—212). These experiments would also require further support, including a comparative assessment of host cell growth rates (as stressed in ref. 47), demonstrating that all signaling molecules between cells and plasmids are orthogonal, etc.

We agree with the reviewer in that all those experiments would be very interesting. In fact, the conclusion of the paper is that a new way of engineering circuits (i.e. those experiments the reviewer mentions) is indeed possible due to the emergence of context-dependency (i.e. the results shown). We thank the reviewer for acknowledging the importance of such future experiments—and many more that are to come. However, that will be the target of future efforts.

2. The authors cite several papers to support the notion that distributed cellular computing is an active field and therefore a viable solution. However, there are significant limitations of this approach, which are not discussed in the manuscript (refs. 30, 32, 46, 47). The authors should add a more nuanced discussion of the drawbacks of distributed computing (e.g., differences in cell-to-cell behavior, orthogonality of signaling molecules, etc.).

Whilst the reviewer is correct in pointing out that there are still limitations to distributed cellular computing as an approach, the target of this manuscript is not a nuanced discussion of these limitations. Instead we prefer to refer the reader to existing work which deals with distributed cellular computing directly. However, the authors do wish to avoid misrepresenting the immediate viability of the distributed cellular computing approach, especially in connection with potential applications of the work in this manuscript. As such the edited Discussion acknowledges that more work in this space would be required in order to mitigate these limitations for an application of this work.

Revised text (Edited Line 285 - 289):

Multicellular distributed computing could potentially enable such a use of chassis as a design parameter in genetic circuits, and has been the target of much recent work which has shown great potential, despite some limitations that must yet be overcome^{30, 32, 46, 47}. Our analysis suggests that multicellular computing approaches may be suited to implement a wide variety of functions, and establishes rational criteria for the selection of cellular chassis in such distributed consortia from a bottom-up design.

3. The authors use the term "non-linear patterns" to describe the idea that the change in behavior for one gate between two different contexts may or may not map onto the change in behavior for another gate between the same two contexts. The authors should provide further justification for using this mathematical term or consider replacing it.

We justify the use of 'nonlinear' with the results shown in Figure 2B, and the caption text. Specifically, we use the term nonlinear because we were unable to identify a linear transformation (through optimisation) which consistently maps gate behaviour between contexts. We agree with the reviewer that our use of the term may cause confusion with readers, and that our justification was lacking. We have addressed this with a revision to make the Figure2B caption text clearer, and have also expanded this with further discussion in the main text.

Revised text (Edited Line 146 - 155):

The issue of inter-context predictions arose as a formidable challenge. For example in Figure 2, an attempt to predict the performance that gates would display in the context *E. coli* DH5 α (pSEVA221), based on gate performance in *E. coli* CC118 λ pir (pSEVA221) failed. The prediction was based on applying an optimised linear transformation to the gates transfer function. No linear transformation that performed consistently well could be found using this procedure, suggesting that a nonlinear transformation may be required. In this case, the optimisation was done using the AmtR-A1 gate, and the predictions tested on other gates in the library. As expected, some of the gates showed a relatively good prediction (good candidates for portability applications), but that was not the case for all of the constructs. Although predictable, gate portability is highlighted as an open problem and contextual dependencies offer a unique opportunity for fine-tuning gate performance, which we carefully analysed as explained next.

Revised Figure 2B Caption:

Nonlinearities made the prediction of gate performance changes between contexts an overarching challenge. Predictions were made for gates in the *E. coli* DH5 α (pSEVA221) context ('Predicted' line), based on their characterisations in *E. coli* CC118 λ pir (pSEVA221) i.e. 'Reference' line. Predictions were made using a transformation matrix found by searching for the optimal linear transformation between the AmtR-A1 gates in each context. The actual characterisation of the gate is shown for comparison ('Measured' line). It can be seen that the optimised linear transformations cannot accurately predict changes in gate behaviour between contexts. In particular, although translations (a linear transformation) in the Input and Output axis appear to be predicted well in some cases (see for example QacR-Q2), more qualitative changes in the shape of the response curve cannot be addressed by this linear transformation (see for example QacR-Q1). All response curves are plotted in RPU-RPU.

Minor Concerns

1. On page 5, line 139 there is a typo – "non-linear patters"

Edited as required. Also, changed to 'nonlinear'.

2. On page 3, line 80-82 – this description of the NOT function should be revised for clarity. The use of double negatives (e.g., "...not negatively regulated...") may be confusing to some readers.

We have revised the description to improve clarity, as follows.

Revised text (Edited Line 88 – 89):

The logic function (NOT or inverter) corresponds to a genetic device that reverses the incoming signal (i.e. output high to input low and vice-versa).

3. From the methods, it appears that the Input is the IPTG regulator and the Output is the YFP reporter, which are quantified in author-defined units of Reference Promoter Units (RPU). These labels are missing completely from Fig. 1, Fig. 2, and Fig. 4. While Fig. 3 does show units of RPU, it does not label the identity of the molecule being quantified (i.e., IPTG, YFP). These labels should be added to all figures and mentioned in the captions.

All axes on all plots of transfer functions are in RPU. With respect to identity of the molecules involved (in plots), part of the rationale for choosing RPU was to obviate the need to identify these – the transfer functions are intended to be indifferent of the inducer and reporter molecules (see Nielsen et al. Science, DOI:10.1126/science.aac7341). Nevertheless, the specifics of the measurements from which we derive RPU is presented in the Methods, and our description of RPU in the main text has been revised for clarity.

Revised text (Edited Lines 95 – 97):

"Relative promoter units (RPU) for both the inputs and outputs of transfer functions were derived from yfp fluorescence measurements, in order to standardise their characterisations."

Revised text (Edited Line 340):

Relative Promoter Unit

Revised caption Figure 4:

Figure 4. Calculation of maximum circuit depth as a result of layering inverters. Based on the compatibility between gates, these were layered within the library in order to evaluate the impact of contextual dependencies on circuit size. For all graphs: x-axis refers to the input and y-axis to the output (both RPU). A. The maximum depth calculated when the computational method is forced to consider all gates carried by the low copy-number plasmid pSEVA221 and hosted by *Escherichia coli* CC118 λ pir, is 3 gates-deep. B. If the algorithm is free to select any plasmid (but still forced to *E. coli* CC118 λ pir), the maximum depth increases to 5. In this scenario, two gates are carried by the medium copy-number plasmid pSEVA231. C. In the last analysis, the calculation used all contextual dependencies, including the variation in host chassis. The maximum number of gates layered increases to 12 (only 5 shown in figure—refer to Supplementary Material for more information). In the sketch shown in the figure, 4 out of the 5 gates were characterised in the strain *Escherichia coli* DH5 α .

4. Page 3, line 95 – the authors should add further clarification on why these specific cell lines were chosen, and why a larger and more diverse library was not used.

The bacteria *E. coli* is the model organism that is most commonly used for synbio fundamental developments. Also, it is the model organism of previous efforts relevant to our work. Therefore, we considered it to be a good choice to begin with. We chose 2 different strains of this species to investigate contextual effects among cell lines that are evolutionary relatively close. In addition to this, we wanted to compare against an organism that is evolutionarily distant to the previous ones. In order to achieve this, we opted for the bacteria *P. putida*, a common workhorse for environmental applications. Moreover, our recent experience (<https://doi.org/10.1111/1462-2920.14544>) suggests that *E. coli* and *P. putida* have very different regulatory dynamics, which makes them ideal for comparing the functioning of regulatory circuits.

Concerning the gate library, we have used one of the largest NOT libraries available (DOIs: 10.1126/science.aac7341, 10.15252/msb.20199401, 10.1038/s41587-020-0468-5) and multiplied its functionality several fold by using 2 different plasmids and 3 different origins of replication per gate.

Altogether, we consider this library to be large and diverse.

Moreover, we make available the code and Methods for analysis, in the hope that if other researchers would like to characterise other/larger libraries, they can do so, and compare/add their results to those presented here. We mentioned in the revised manuscript this potential for scalability with respect to the analysis of other libraries.

Revised text (Edited Line 113 – 118):

While the performance changes abruptly in some cases (e.g., in contexts 3 and 4), it did not change significantly in others (e.g. in contexts 5 and 6). The codes and methods used for this analysis are made available to encourage extensions to this work and its application to other data sets (see Methods). Overall, our analysis suggests that contextual dependencies act as a hidden layer of parameters that must be carefully considered to achieve a predictable logic gate design—an issue which has been traditionally overlooked.

Revised text (Edited Line 293 – 295):

Although in this work we have considered backbones and strains as the contextual dependencies, this library could be extended by adding other contextual parameters such as other promoters, context-specific genetic parts, or substrates.

Revised text (Edited Line 105 – 107):

Regarding the chassis, we used two *Escherichia coli* (DH5 α and 10090) strains that are evolutionary relatively close and one *Pseudomonas putida* (KT2440) strain that is an evolutionary distant host to the other two.

5. In multiple places (e.g., page 2, line 49-50; page 4, line 108) the authors claim that the role of context in genetic circuits is traditionally overlooked. They should add citations to support this claim.

We do not intend to state that role of context not known to the community. However, we argue that there is a lack of work (particularly in logic gate design/characterisation) which attempts to give strategies that co opt these contextual effects, that is, contextual effects are overwhelmingly regarded as antagonistic. We aim to frame contextual effects in a different light, as expanding the design space available for synthetic biologists. Paragraph 2 of Introduction has been revised to make our position clearer.

Revised text (Edited Lines 47 – 63):

A fundamental challenge for the design of robust synthetic circuits, which underpins this work, is the oversimplified model that assumes DNA elements (i.e., gates) alone explain the performance of genetic circuits. Based on this assumption, the host chassis (the cell that receives a specific genetic construct) is generally ignored and the interplay of a genetic circuit with the host context is most often overlooked in the bottom-up design process—an issue that has been identified essential for the predictability of synthetic biology devices¹². Our results here suggest that, rather than being antagonistic, incorporating context into the design of biological circuits can actually provide advantages by enlarging the available design space. Both the burden imposed by synthetic constructs on the host^{13, 14} and the impact of context on genetic activity¹⁵, have phenotypic implications that cannot be predicted from a gene-centric standpoint. Recently, the term host-awareness^{16, 17} has been coined to bring attention to this problem, which is at the core of the lack of part interoperability¹⁸ (i.e., parts that show similar performance in different host contexts). Here, we propose to utilise a strategy that is inspired by nature and includes context as a parameter in the design of optimal genetic circuits.

6. The gate-compatibility score seems useful because it could be applied to any Boolean gate in any system, and is particularly useful for biological gates because HIGH/LOW values are not universally defined. I'm not sure why this is buried in the supplement; the authors could consider bringing this formulation to the main text with explanation.

We agree that the gate-compatibility score is a useful metric. We have revised the Methods to include the formula under its own heading in Methods. We have also added a brief discussion in the main text as part of the discussion of Figure 3A.

Edited Methods (a new subtitle):

Compatibility Scoring

$$\min \left(\ln \left(\frac{IL_B}{OL_A} \right), \ln \left(\frac{OH_A}{IH_B} \right) \right)$$

For a pair of operational inverters, A and B, their compatibility score was defined as:

A positive score indicates that a high(low) output from A will also be interpreted as high(low) by B, because IL(IH) of B is greater(less) than OL(OH) of A. From this we imply that inverter A can be connected as input to inverter B, if and only if their compatibility score is positive.

Revised text (Edited Line 166 – 173):

In order to tackle this contextual issue, we first scored the matching of all the gate pairs in the library according to their input and output thresholds (Figure 3A). The inclusion of the input thresholds in the output ones defines a pair as “compatible”. The extent of the inclusion is used to compute a compatibility score (formula presented in Methods), which is positive if the pairing is compatible and negative otherwise. This metric permits the comparison of all available compatible pairings and potentially informs design decisions. That is, under this framework, a design consisting of pairings with larger positive scores should be preferred over designs with comparatively smaller or negative scores. With this in mind, scoring of all pairings in a library may indicate the overall quality of a gate library and of the circuits produced thereof.

7. Page 6 line 181-182 – The authors mention that when considering combining gates from different contexts, the total number of compatible gates (X) increases, but so does the number of possible gates (Y). Importantly, they mention that the percent of total possible combinations remains constant (i.e., normalize X/Y). This normalized percent should be represented in the figure for all three panels (i.e., pAN only, Any plasmid, Any Context) to avoid misleading the readers.

The reviewer raises an important issue and we thank him/her for helping identify a typo in the manuscript concerning this %. Upon further inspection, we have also revised our method for computing these percentages. These results, along with the method, are provided in full in the supplementary material in Table S24.

Upon reflection, we consider that the percentage increase in the number of compatible gates, as contextual parameters are relaxed, is an important metric provided in our analysis. Therefore, we have extended the discussion on this in the main text, as part of the discussion around Figure 3. We have also revised the caption of Figure 3C to reflect these numbers. However, we did not change the Figure itself, since we want it to highlight increase in the total number (what it does already), and leave % increase to the description in text.

Revised text (Edited Lines 181 – 192):

As a general trend, relaxation of the contextual parameters of backbone and host results in an increase in compatible pairings. More importantly, this increase is made up of not only new pairings within the additional contexts, but also additional pairings between gates in different contexts. In the example shown in Figure 3C, 85

pairings are possible in DH5 α without using two different backbones, 67 with pAN (Figure 3C, left) and 18 with pSeva221 (Supplementary Figure S3), whereas 203 pairings are possible when allowing mixing of these backbones (Figure 3C, middle). Thus, compatible pairings in the library increased ~240% as a result of incorporating connections between gates with different backbones. A similar jump of ~240% is observed when incorporating connections between gates with different hosts in addition to different backbones (Figure 3C, right). We conclude that consideration of backbone and host as a design parameter results in a more flexible, and reconfigurable, library with the ability to include dynamics that are not captured by just DNA sequences e.g., the copy number of circuits (thus their burden to the cell).

Revised caption Figure 3:

(...) The freedom to use both backbone and strain as a design parameter yields the most compatible pairs at 697. Further, from 19 functional NOT gates, with a possible 320 pairings between them, 198 of these (61.8%) can be realised by allowing different backbones and strains to be utilised.

8. In Fig. 3C, why was pAN chosen as the single context to isolate in the main figures? It appears that from the supplemental figures, all contexts were tested, and that some imposed almost no compatibility constraints (e.g., DH5 α host with the pSeva221 backbone.) Further, why was the DH5 α host fixed for the "Any plasmid" table? Were other hosts not tested in this same manner? For completeness, the authors should also fix each plasmid and enable connections between different host cells. Were these specific isolations/contexts chosen simply to make the data show the trend that releasing constraints on gate context increases the number of total compatible gates?

All 7 combinations show the same trend. The revised supplementary material includes the relevant plots which support this.

Note: out of 11 possible contexts, we have tested 7. The reasons for this (e.g. origin of replication of plasmid pAN not being suitable for *P. putida*) are outlined in the text.

9. The compatibility score is a continuous value and is reported as such in the supplemental figures. In Fig. 3, these values are binarized. How was this threshold chosen? If it was that positive scores are defined as compatible, how did you differentiate gates that are very positive from gates that are very close to 0? It is easy to imagine that with biological variance the latter gates would be subject to failing.

Thresholds in Figure 3 are chosen in the manner the reviewer suggests, as is presented in Methods, under the heading of 'Calculating compatibility'. The reviewer is correct that we would expect gate combinations with low scores would be more likely to fail than those with high scores. The score itself provides a way to differentiate between these two cases.

Reviewer #2

This paper addresses the important topic of host context dependence in genetic circuit elements. Particularly, genetic circuit elements described using Boolean algebra that are intended to be composable into large functions. It focuses on NOT gates used in 7 bacterial based contexts.

This summarizes well the manuscript.

The key elements that need to be demonstrated in my opinion for this type of work are:

1. Reproducibility – this appears to be fine in this case from an experimental standpoint. In particular, a strength is the fact that their approach uses existing gate "architectures" from the "Cello" work. Naturally, if the author's

work is also able to be automated, details on an automation framework to provide this analysis would be great. This would include capturing the protocols in frameworks like Autoprotocol, Aquarium, [Protocols.io](https://protocols.io), etc. If automation is used, scripts for robots (e.g. PyHamilton scripts or even CSV picklist files), would be great.

While we did not make use of an automation framework/tool for the experiments in this present work, we agree with the reviewer in that this would be a natural next step for future work. As far as reproducibility is concern, we provide the code used for the analysis of the experimental data and the data itself.

2. Performance metrics – an interesting phenomenon of this work is that the NOT gates can have different functions in different contexts. A way to illustrate which Boolean algebraic function the circuits can implement could be shown in a “cosine similarity” style analysis like in [“Large-scale design of robust genetic circuits with multiple inputs and outputs for mammalian cells” – Wilson Wong]. The provided data fitting and compatibility analysis equations are adequate for this work.

Certainly, an analysis made for more complex gates might benefit from the similarity measure the reviewer mentions. However, as the reviewer also notes, we consider that our fitting and thresholding operations are an adequate measure for the NOT gates we work with here. In fact, using NOT gates was a conscious choice, made in order to keep this analysis as simple as possible.

3. Standardized measurement – It would have been nice to see something like MEFLs [“Quantification of bacterial fluorescence using independent calibrants” – Jacob Beal] used in the measurement work but the authors to their credit do work with a version of the standardized fluorescence measurements.

We appreciate the reviewer’s interest for standardisation. As the reviewer points out, we use standard measurements, although not MEFLs (but RPU). We opted for the calculation of RPU because previous efforts in circuit design use similar metrics (e.g. [10.1126/science.aac7341](https://doi.org/10.1126/science.aac7341)). We provide the experimental data, along with all other information (e.g. code), so that future efforts on automation could use our RPU or calculate MEFLs.

As far as standardisation is concerned, we used SEVA vectors for the implementation and SBOL descriptions (as requested by the reviewer in another comment)—along with RPU for measurements.

4. Availability of the genetic information and associated data – it is not clear that this work deposited the material in a location that can be retrieved. A suggestion is to create SBOL files for this work and store them in a public SynBioHub instance. As for the genetic material, it would be nice to know how to get these (e.g. Addgene). If those resources as not available to the authors, any proposal for how to accomplish this would be appreciated.

We agree that the availability of SBOL data would enhance the work. SBOL files have been deposited in the associated github and figshare repository, where the experimental data was located. We have specified this in the revised manuscript.

About the physical genetic constructs, they are being included at the SEVA bank at <http://seva-plasmids.com>. Once their specification is uploaded (which will happen during the next update) the plasmids will be ready for distribution—for free for research purposes

Revised text (Edited Lines 399 – 405):

New section - Availability of data, genetic material and supporting software

The flow cytometry data used for analysis in this study is available as a figshare repository at <https://data.ncl.ac.uk/ndownloader/articles/12073479/versions/1>. This repository also contains SBOL files for the genetic constructs used in the study. The constructs themselves are retained at SEVA bank at CNB-CSIC, Madrid, Spain and ready for distribution for research purposes. The Python package used to perform all the

analysis presented, the preprocessing of the raw cytometry data, and to generate the Figures shown here, is made available at <https://github.com/lgrozinger/pyolin>.

5. A method for others to use this data – while the tables and diagrams are great, this work really would lend itself toward a very simple design tool. The user could either enter the desired transfer function (using the functions specified in the paper) and/or host context. The tool then would present the viable options solely based on the data already calculated.

In the absence of a program, a clever way to use the heat maps along with transfer functions would be nice.

As it currently stands, there is no clearly accessible mechanism to make use of this data programmatically. This is a criticism of almost all work in this space. However, this work is very close to addressing this.

Python code for the data analysis work is provided (a strength), perhaps this could be improved to provide some of this design functionality.

As stated before, we agree with the reviewer in that such a design tool would be a natural next step in this direction. However, it was not the goal of this work. We have noted in the supplementary material, however, that the Python package is capable of generating minimal 'UCF' files for consumption by the 'Cello' automated design framework (Nielsen et al. Science, DOI:10.1126/science.aac7341). In future work, the Python package provided, may well be further developed into a design tool. As said, this is beyond the scope of the current manuscript.

6. Design rules that formally encode the context data discovered in this work – this work does address some of these elements with the compatibility matrices. However, these are not formalized. Again, if the authors were so inclined these can be written as a formalized set of mathematical rules related to composition. These rules could be written in python and used for a circuit design tool.

We believe the response above also addresses this comment. The Python code which is used to calculate compatibility (rules, thresholds, etc.) is available at the associated github repository, and is capable of generating CSV files of numerical data, as well as the 'UCF' files already mentioned above. Altogether, this could be used for developing a circuit design tool, which is beyond the scope here.

Another aspect that is not clear, is the role of other effects (e.g. toxicity) on the ability of the circuits to be composed. There are examples of the number of layers available but it is unclear how realistic these are. That might not be the point, but in the case of a paper that is trying to take broadly applicable ideas (gate composition) and make them realistic (host-context), ideas like this are a noticeable omission.

It is indeed most interesting to discuss this issue, which was overlooked in the original manuscript and now included in the revised text. In fact, five of the genetic inverters were not functional when cloned into a high-copy plasmid (pSEVA251), which suggests there is a toxicity effect or an overload of cellular resources. With an increasing interest in studying resource allocation dynamics, this discussion points at an important intersection of context-dependency and cellular economy.

Revised text (Edited Line 250 – 254; In Discussion):

Therefore, genetic logic gates are exposed to contextual dependencies that influence their phenotypic behaviour. In the extreme case, these constraints can even result in inviability of the cellular host, for example, during the conversion to pSEVA broad host range backbones, five of the genetic inverters were not functional when cloned into a high-copy plasmid (pSEVA251), perhaps due to overload in the allocation of cellular resources resulting in toxicity.

General Comments:

Why stop at 7 bacterial contexts or the fixed NOR gate architecture? It might be nice to add a couple of others (contexts like *Geobacter* or NOR gates using other promoter/repressor combinations) that are chosen AFTER you have done the initial analysis to confirm or reject predictions or to see how broadly applicable the lessons learned are.

This is indeed a very relevant issue. The reason why the inverter architecture is picked is that NOT gates (so does NOR gates) are building blocks of all Boolean logic operations, hence the aim here was to test such a universal architecture. We chose 2 different strains of this *E. coli* to investigate contextual effects among cell lines that are evolutionary relatively close. In addition to this, we wanted to compare against an organism that is evolutionarily distant to the *E. coli*. In order to achieve this, we opted for the bacteria *P. putida*, a common workhorse for environmental applications. Moreover, our recent experience (<https://doi.org/10.1111/1462-2920.14544>) suggests that *E. coli* and *P. putida* have very different regulatory dynamics, which makes them ideal for comparing the functioning of regulatory circuits. It could be interesting to extend the work among different bacteria and gate architectures, however, this would require additional experiments and so would be the target of future work. The Python package is general enough to perform the same analysis presented here for these future experiments. As it stands, we believe that the diversity of the library and bacterial strains used here makes our analysis of interest and suggests that the broad application of our conclusions may be justified.

I did not see the transfer functions for all 135 gates anywhere including the supplemental. This would be valuable even in the form of spreadsheet data.

As suggested, the parameters of the 135 transfer functions have been made available in the supplement in the form of a table. This is now mentioned in the revised text.

Revised text (Edited line 360 -361):

Fits were obtained for all gates presented in this paper. The fitted parameters are shown in tables in the Supplementary Materials.

Again the gate compatibility analysis screams out for a small tool. It is not clear from the supplemental what the python work includes. In general, it could be made more clear in the supplemental examples of what the software can do.

The documentation available in the github repository has been expanded to clarify the capabilities of the Python software. We have expanded its description in such a way that any future effort on generating a design tool could build on the existing package.

A natural next step now would be to introduce genetic elements that help to normalize the performance across context. Some discussion of this would be ideal in the manuscript.

This is indeed an interesting way to look at the portability issue i.e. the elimination of contextual dependencies. However, the paper aims at highlighting the opposite approach: contextual dependencies should inform about the choice of DNA parts. Our results suggest that the latter would allow for more complex designs, since it does not require any extra synthetic device to inactivate dependencies and achieve normalization (thus increasing burden). Furthermore, we advocate for moving into a more bio-friendly way of engineering, instead of forcing synthetic functions into environments that may not be suitable for that. In fact, we find that in natural systems. For instance, *E. coli* solves energy requirements through the EMP metabolic pathway, while *P. putida* does it via the ED pathway. The function is the same: glucose as input and energy as output, but the circuitry is not normalized. Rather, it depends on the context. We added a brief discussion on this into the revised manuscript.

Revised text (Edited Lines 54 – 63):

Both the burden imposed by synthetic constructs on the host^{13, 14} and the impact of context on genetic activity¹⁵, have phenotypic implications that cannot be predicted from a gene-centric standpoint. A common strategy seen in nature is to achieve a similar outcome using a different pathway in different organisms, rather than normalising

pathways across all organism. For instance, *E. coli* solves energy requirements through the EMP metabolic pathway, while *P. putida* does it via the ED pathway. The function is the same: glucose as input and energy as output, but the circuitry is not normalised. Rather, it depends on the context. Here, we propose to utilise a strategy that is inspired by the natural one, and includes context as a parameter in the design of optimal genetic circuits. Recently, the term host-awareness^{16, 17} has been coined to bring attention to this problem, which is at the core of the lack of part interoperability¹⁸ (i.e., parts that show similar performance in different host contexts). Here, we propose to utilise a strategy that is inspired by the natural one, and includes context as parameter in the design of optimal genetic circuits.

I would expect the supplemental to have all of the gate information related to Figure 2 along with the actual numerical scores. Perhaps this is covered by the Python package?

Yes, the Python package can be used to generate all of the analysis presented. In addition, tables of numeric data, including the fitness scores mentioned, have been added to the supplementary material.

In general, the supplemental material should be expanded with more data if nothing else for reproducibility checks across labs interested in this work.

This has been done, as discussed above.

In general, layering inverters is not particularly useful UNLESS you can produce different inverter behaviors with an odd number of inverters not possible by individual inverters or other collections of odd inverters. Understanding if new behaviors can be introduced with the layered gates would have been interesting.

The goal of the work is not to layer inverters per se. Rather, the manuscript is about using this layering to showcase the increase in compatibility due to contextual dependencies, in as much as the maximal layering of NOT gates represents an upper bound on the depth of any circuit built with the library. The intended utility of the layering depths we report has been made clearer in the revised manuscript. We indeed “produced different inverter behaviours” (sic) since a library of 20 functions was turned into a library of 135 functions (and the same DNA parts). This was not done by coupling an odd number of inverters, but by changing dependencies. Future efforts will revolve around the use of those 135 functions (or even more if other contextual dependencies are added according to the methods suggested) for engineering more complex information-processing devices than NOT gates.

Revised text (Edited Lines 211 – 228):

To begin to address this question, we carried out computations in order to identify the maximum circuit depth (i.e., number of layers³⁵) that could be achieved by connecting gates within our library, and assessed the impact of contextual effects in such a chain (Figure 4). First, when considering all gates in the same context, with backbone pSEVA221 and hosted by *E. coli* CC118 λ pir, the maximum depth was 3 (Figure 4A). That is, there are 3 gates that can be connected consecutively while maintaining the correct logic output (i.e., logic values 0/1 are effectively transmitted from beginning to end). Every other valid configuration will result in fewer (or the same) number of layers. We find that increasing the number of contexts available can significantly increase the maximum depth computed by the search algorithm. As shown in Figure 4B, allowing another context by including gates characterised with any backbone (but still hosted by *E. coli* CC118 λ pir) increases the maximum depth to 5. This can be further improved upon by allowing freedom in the choice of host, for a total of 7 contexts (Figure 4C). In this case, the computed maximum depth is 11, far beyond the current state-of-the-art for synthetic circuitry¹. Of course, this maximum circuit depth is a hard upper bound on the depth of any circuit that could be constructed using the library, but does not guarantee that this depth can be achieved in a circuit that does not simply layer inverters. However, the increasing depth we observe as contextual parameters are relaxed suggests that there is potential for deeper circuits with context as a design parameter than without. Libraries of genetic gates which are allowed to be placed in multiple contexts appear to be less restrictive than their single-context counterparts, and could potentially permit a broader range of more complex circuit designs.

Reviewer #1 (Remarks to the Author):

In the revised manuscript, the authors sufficiently addressed most minor concerns regarding edits to the text and clarity of the manuscript. However, the authors did not address my previous concerns regarding significance. The arguments posed against these concerns are unpersuasive in my view, and therefore the same concerns remain.

First, in the manner presented in Figure 4, these results are simulations (i.e., an approximate imitation of a system), albeit very simple simulations. The authors clearly propose a system by depicting a host cell, plasmid backbone, and specific sequence(s) of inverters, and then calculate a theoretical result (maximum circuit depth). This is a theoretical approximation of a complex system, and in the case of panel C, a distributed computing system, which may or may not behave as the authors claim when realized in practice. The authors go beyond assessing gate compatibility here; the authors claim that it would be possible to combine these gates in a very specific way, so indeed, these claims should be experimentally validated. Models that lack validation are not considered predictive and would have limited value.

Second, the notion that not calling something a simulation means that it doesn't need to be supported with experimental evidence doesn't make any sense. Even if the results in Fig. 4 are not called "simulations", the authors use the analysis for the final figure of their paper to claim that their results could lead to a circuit-depth "far beyond the current state-of-the-art for synthetic circuitry". For this claim to carry any weight, the authors should do something to make the reader believe this is even practical. It seems the best way to do this would be with experimental proof. The problem, it appears, is that such an experiment would require the authors to implement a highly complex distributed computing system, which may or may not be practical.

Indeed, if swapping out contexts is overall better (e.g., easier, cheaper, enabling, etc.) than swapping out for new components – as is the central hypothesis of the paper – then this story may elicit broad interest. Of course, this hypothesis is weakened if their solution is more resource-efficient, but much more difficult to implement than gate-swapping. But it is only a hypothesis until the authors prove this claim experimentally. In its current form, the paper only recreates existing NOT gates in isolated contexts.

Reviewer #2 (Remarks to the Author):

The following is an assessment of the revised manuscript "Contextual dependencies expand the re-usability of genetic inverters".

The addition of the following elements make the paper stronger and address many of my initial concerns:

1. Python analysis code and associated data. Expanded descriptions and documentation.
2. SBOL files – In GitHub
3. SEVA vector usage – in SEVA repo
4. Short discussion on other context effects (e.g. toxicity)
5. All 135 gate transfer functions have been added to the supplemental text

Overall, this paper now exceeds the amount of data analysis, documentation, and "availability" of similar papers in this domain. The authors have made a good faith effort to address many of my concerns.

There are two points that I still feel limit the potential impact of this paper:

1. Hesitation (on inability) to create any design software to accompany this. The authors say this is out of scope and that it can be done by others. Perhaps it will be. I suspect however that this work is not unambiguous enough either in its protocols or software workflow to prevent multiple

labs from getting multiple results using their own design tools built on their assumptions. This would be addressed with a centralized, "official" tool. This should not be cause for the paper to be rejected but nevertheless is a missed opportunity in my opinion on the part of the authors.

2. Promoting a narrowing of the work (specific context, specific gate types). This is likely a difference in philosophies. The authors do not seem to be as motivated to use the context-dependent phenomenon they observe to help move toward context independence. I agree that actually doing this experimentally is outside the scope of the work. However, if the idea is that for each context, an analysis of this magnitude need be performed each time, much of the engineering approaches in synthetic biology simply won't be applicable.

In any event, the authors' revisions are substantive and make the paper worthy of publication.

Response to reviewers' comments to manuscript NCOMMS-20-28880

(answers in blue)

Reviewer #1

In the revised manuscript, the authors sufficiently addressed most minor concerns regarding edits to the text and clarity of the manuscript. However, the authors did not address my previous concerns regarding significance. The arguments posed against these concerns are unpersuasive in my view, and therefore the same concerns remain.

First, in the manner presented in Figure 4, these results are simulations (i.e., an approximate imitation of a system), albeit very simple simulations. The authors clearly propose a system by depicting a host cell, plasmid backbone, and specific sequence(s) of inverters, and then calculate a theoretical result (maximum circuit depth). This is a theoretical approximation of a complex system, and in the case of panel C, a distributed computing system, which may or may not behave as the authors claim when realized in practice. The authors go beyond assessing gate compatibility here; the authors claim that it would be possible to combine these gates in a very specific way, so indeed, these claims should be experimentally validated. Models that lack validation are not considered predictive and would have limited value.

Second, the notion that not calling something a simulation means that it doesn't need to be supported with experimental evidence doesn't make any sense. Even if the results in Fig. 4 are not called "simulations", the authors use the analysis for the final figure of their paper to claim that their results could lead to a circuit-depth "far beyond the current state-of-the-art for synthetic circuitry". For this claim to carry any weight, the authors should do something to make the reader believe this is even practical. It seems the best way to do this would be with experimental proof. The problem, it appears, is that such an experiment would require the authors to implement a highly complex distributed computing system, which may or may not be practical.

Indeed, if swapping out contexts is overall better (e.g., easier, cheaper, enabling, etc.) than swapping out for new components – as is the central hypothesis of the paper – then this story may elicit broad interest. Of course, this hypothesis is weakened if their solution is more resource-efficient, but much more difficult to implement than gate-swapping. But it is only a hypothesis until the authors prove this claim experimentally. In its current form, the paper only recreates existing NOT gates in isolated contexts.

We thank the reviewer for the comments. While we do understand that further experiments would be of interest, we argue these would not make the message of our results any stronger. Furthermore, the experiments suggested are complex, and beyond the scope of the current manuscript. We do not argue that swapping out contexts is "better" (as the reviewer points out) than swapping out for new components—rather, swapping out contexts adds an extra layer on the performance that affects compatibility. That is what we tested experimentally.

However, we also understand that any future attempt on building a multicellular context-dependent circuit would need some guidance in order to make the most of the present results. We have expanded the discussion to this end, providing rules to be followed by experimentalists. We added the next:

Consideration of different contextual dependencies was found to increase the theoretical maximum circuit depth from 3 to 12, as shown in Figure 4. Practical implementations of such context-dependent designs would comply with the following three rules. Firstly, designs necessarily rely on multicellular distributed computing approach in order to connect logic gates in different hosts. These connections could be established by using orthogonal

quorum sensing (QS) systems. In order to prevent the number of orthogonal QS systems from being a limiting factor, paths should be selected that minimise these inter-host connections—therefore maximising the number of gates per host. Secondly, linking gates inside a host requires using repressor molecules for signalling, which must also be orthogonal to ensure correct gate operation. Since the library of repressors would also be limited, the adoption of multicellular approaches offers an important advantage: the reusability of parts i.e. a repressor that is used in one host may be reused in another. This second rule may be used to distribute circuit burden across different strains. Finally, plasmid backbones can coexist inside the same host as long as their origins of replication are different—otherwise, some plasmids may be lost during the process. Altogether, these guidelines establish a rational criteria for the selection of context-dependent circuitry components from a bottom-up design.

Reviewer #2

The following is an assessment of the revised manuscript “Contextual dependencies expand the re-usability of genetic inverters”.

The addition of the following elements make the paper stronger and address many of my initial concerns:

1. Python analysis code and associated data. Expanded descriptions and documentation.
2. SBOL files – In GitHub
3. SEVA vector usage – in SEVA repo
4. Short discussion on other context effects (e.g. toxicity)
5. All 135 gate transfer functions have been added to the supplemental text

Overall, this paper now exceeds the amount of data analysis, documentation, and “availability” of similar papers in this domain. The authors have made a good faith effort to address many of my concerns.

There are two points that I still feel limit the potential impact of this paper:

1. Hesitation (on inability) to create any design software to accompany this. The authors say this is out of scope and that it can be done by others. Perhaps it will be. I suspect however that this work is not unambiguous enough either in its protocols or software workflow to prevent multiple labs from getting multiple results using their own design tools built on their assumptions. This would be addressed with a centralized, “official” tool. This should not be cause for the paper to be rejected but nevertheless is a missed opportunity in my opinion on the part of the authors.
2. Promoting a narrowing of the work (specific context, specific gate types). This is likely a difference in philosophies. The authors do not seem to be as motivated to use the context-dependent phenomenon they observe to help move toward context independence. I agree that actually doing this experimentally is outside the scope of the work. However, if the idea is that for each context, an analysis of this magnitude need be performed each time, much of the engineering approaches in synthetic biology simply won't be applicable.

In any event, the authors' revisions are substantive and make the paper worthy of publication.

We thank the reviewer for the positive feedback/comments. Concerning point 1, we believe that the ability of the provided package of generating UCF files (with context sensitivity) would be enough to provide a good baseline for using Cello as the “official” tool. On point 2, we would just like to mention that the manuscript focusses on

context dependency as a way to make the most of the flexibility of biological systems; but we agree that context independent circuits is a major goal that may benefit from our results.